# Effect of Systemic Steroid Use for Immune-Related Adverse Events in Patients with Non-Small Cell Lung Cancer Receiving PD-1 Blockade Drugs

**DOI:** 10.3390/jcm10163744

**Published:** 2021-08-23

**Authors:** Atsuto Mouri, Kyoichi Kaira, Ou Yamaguchi, Kousuke Hashimoto, Yu Miura, Ayako Shiono, Shun Shinomiya, Hisao Imai, Kunihiko Kobayashi, Hiroshi Kagamu

**Affiliations:** Comprehensive Cancer Center, International Medical Center, Department of Respiratory Medicine, Saitama Medical University, 1397-1 Yamane, Hidaka, Saitama 350-1298, Japan; kkaira1970@yahoo.co.jp (K.K.); ouyamagu@saitama-med.ac.jp (O.Y.); hkosuke@saitama-med.ac.jp (K.H.); you_mi@saitama-med.ac.jp (Y.M.); respiratory@hotmail.co.jp (A.S.); shinomiso4038@gmail.com (S.S.); hisao725@saitama-med.ac.jp (H.I.); kobakuni@saitama-med.ac.jp (K.K.); kagamu19@saitama-med.ac.jp (H.K.)

**Keywords:** steroid, immune-check point inhibitor, immune-related adverse events, non-small cell carcinoma, efficacy

## Abstract

Objectives: Programmed death-1(PD-1)/programmed death ligand-1 (PD-L1) antibodies have clinical benefits for cancer patients facing immune-related adverse events (irAEs). However, the effect of steroid use on the prognosis of patients with non-small cell lung cancer (NSCLC) receiving PD-1 blockade remains unclear. Methods: NSCLC patients with complete response (CR)/partial response (PR) or stable disease (SD)/not evaluable (NE) status plus progression-free survival (PFS) of 180 days after PD-1 blockade from December 2015 to December 2018 were retrospectively registered in our study and were divided into two groups: those with and without systemic steroid use for irAEs. Results: In total, 126 patients who had benefitted from PD-1 blockade were enrolled in our study; among them, 44 received systemic steroids for irAEs, and 82 had no adverse events or, if they did, did not receive systemic steroids. Among the 44 patients requiring steroids, interstitial lung disease (ILD), adrenal insufficiency, diarrhea, and liver dysfunction were observed in 19, 9, 4, and 4 patients, respectively. More side effects were observed in the group treated by steroids. The median PFS and overall survival (OS) in patients with and without systemic steroid use were 11.7 and 16.0 months (*p* < 0.037) and 35.0 and 41.0 months (*p* < 0.28), respectively. In univariate and multivariate analyses of survival, systemic steroid treatment for irAEs was significantly associated with PFS. The occurrence of ILD, adrenal insufficiency, and fever was significant in patients who used systemic steroids for irAEs. Conclusions: Patients administered systemic steroids for irAEs due to PD-1 blockade treatment exhibited shorter PFS than those who were not. Systemic steroids might affect survival after PD-1 blockade even for patients who once acquired its clinical benefit.

## 1. Introduction

Immune checkpoint inhibitors (ICIs), such as programmed death-1(PD-1)/programmed death ligand-1 (PD-L1) antibodies, have been recognized as standard care for patients with advanced non-small cell carcinoma (NSCLC). Several phase III trials in patients with advanced NSCLC have shown the revolutionary efficacy and moderate safety of ICI monotherapy or combined platinum-based regimens with ICIs [1,2,3,4]. However, immune-related adverse events (irAEs) frequently occur in patients who exhibit a therapeutic response to ICI treatment [5]. Steroids and immunosuppressive agents have been identified as the most important countermeasures for treating irAEs. In particular, careful attention to the management of irAEs should be indispensable in patients with lung cancer, melanoma, and renal cancer during combined treatment with ICIs, such as PD-1 blockade and cytotoxic T-lymphocyte-associated protein (CTLA-4) monoclonal antibodies [6,7,8]. The administration of systemic steroids for any irAEs should be considered according to the common terminology criteria for adverse events (CTCAE) grade. Little is known about the detailed mechanism systemic steroids and their effect on irAEs; however, steroid therapy plays a crucial role in the management of irAEs. Systemic steroids act as anti-inflammatory agents. ATII cells are the alveolus defenders. It has been shown that glucocorticoid exposure decreases the transcription of pro-inflammatory cytokine genes, such as tumor necrosis factor-α, interleukin (IL)-1β, and IL-6, in human mononuclear cells [9]. Inflammatory cytokine depletion may also affect type II alveolar epithelial cells that function in alveolar defense. Generally, systemic steroids have been suggested to reduce phagocytic activity; inhibit leukocyte migration; reduce interleukin (IL)-1, IL-6, IL-8, IL-12, and tumor necrosis factor-α; and impair T-cell activity and differentiation [10]. However, the appropriate dosage of steroids for irAE management remain unclear.

Arbour et al. reported the effect of baseline corticosteroids on the efficacy of PD-1 blockade in patients with NSCLC [11]. In their study, progression-free survival (PFS) and overall survival (OS) in patients receiving baseline steroid doses of ≥10 mg/day at ICI initiation were significantly shorter than those in patients receiving <10 mg/day [11]. Although patients with brain metastases and poor Eastern Cooperative Oncology Group performance status were more common in the group prescribed systemic steroids, systemic steroids on baseline treatment may cause negative effect on ICI therapy. Giovanni et al. showed that the PFS and OS were significantly longer in the early phase (within 28 days) in patients not using prednisone ≥ 10 mg post ICI initiation than in those using prednisone [12]. Considering the results of these reports, ≥10 mg systemic steroids before and in the early phase post ICI administration have a negative effect on therapeutic efficacy and outcome. If severe irAEs are observed during treatment with ICIs, systemic steroids may be required for recovery after withdrawal of ICIs. However, whether the induction of systemic steroids after cessation of ICIs could affect efficacy and survival remains unknown. A recent systematic review described that disease control (PR/SD) and progression-free status at 24 weeks (180 days) significantly predicted OS in patients with NSCLC treated with PD-1 blockade through landmark analysis [13]. The results of this review suggest that patients who benefitted from PD-1 blockade are expected to have prolonged OS. If systemic steroids were administered (after cessation of PD-1 blockade) to patients previously benefitting from PD-1 blockade agents, such as PFS ≥ 180 days or PR, then they may reverse the benefits of PD-1 blockade; however, very little is known about this effect.

On the basis of this background, we compared patients who received systemic steroids for the development of irAEs with those who did not develop irAEs or did not receive steroids for irAEs in patients who had achieved a certain response to ICIs therapy. The aim of this study was to elucidate whether systemic steroids administered to manage irAEs affect the survival of PD-1 blockade in patients with metastatic or recurrent NSCLC.

## 2. Methods

### 2.1. Study Design and Subjects

We retrospectively identified patients with advanced or unresectable NSCLC who received nivolumab, pembrolizumab, or atezolizumab as monotherapy at the Comprehensive Cancer Center, International Medical Center, Saitama University Hospital, Japan, between January 2016 and December 2018. To extract only patients who gained clinical benefit from PD-1 blockade treatment, the patients with stable disease (SD) (or not evaluable (NE)) status plus PFS < 180 days or progressive disease (PD) as the best response were excluded. Patients with complete response (CR), partial response (PR), or PFS ≥ 180 days with SD or NE status after ICI monotherapy were eligible and divided into two groups: with and without systemic steroid use. Systemic steroids for managing irAEs were administered at a dose greater than that of prednisone, equivalent of 10 mg/day, and for a period of more than 2 weeks. This study was approved by the institutional ethics committee of the International Medical Center, Saitama Medical University (10 June 2020). The requirement for written informed consent was waived by the ethics committee of Saitama Medical University because of the retrospective nature of the study.

### 2.2. Treatment and Adverse Events

Pembrolizumab, nivolumab, and atezolizumab were intravenously administered: 3 mg/kg or 240 mg/day every 2 weeks, 200 mg/day every 3 weeks, and 1200 mg/day every 3 weeks, respectively. Complete blood cell count, differential count, routine chemistry measurements, physical examination, and toxicity levels were evaluated through medical examination by physicians. Toxicities were graded according to the CTCAE version 5.0. In accordance with the judgment of each physician, ICI treatment was repeated until disease progression, appearance of severe toxicity, or patient’s refusal for treatment.

### 2.3. Assessment for Clinical Data

We performed computed tomography (CT) or positron emission tomography-computed tomography (PET-CT) imaging before PD-1 blockade treatment for all patients as the baseline tumor assessment. To assess response evaluation or to investigate the cause of the inscrutable findings, CT, magnetic resonance imaging, or PET-CT imaging was performed. Tumor response was evaluated according to the Response Evaluation Criteria in Solid Tumors (RECIST) criteria ver. 1.1 [14]. The best response, estimated from the difference in the maximum tumor diameter from baseline and the maximum shrinkage, was recorded. In this study, patients who achieved CR or PR as the best response were included and defined as responders; patients who obtained SD or NE as the best response with PFS > 180 days were included as the target population. In proportion to the RECIST criteria, ver. 1.1 [14], CR was defined as the disappearance of all target lesions, PR was defined as a decrease in the sum of target lesion diameters by ≥30% compared with that at baseline, PD was defined as an increase of ≥20% in the sum of target lesion diameters compared with the smallest sum during the treatment period, and SD was defined as not fulfilling the standard of PR or PD. PFS was defined as the day from the start of ICI therapy to disease progression or death by any cause. OS was defined as the day from the start of ICI therapy to death due to any event or the last contact.

### 2.4. Statistical Analysis

Statistical analyses were performed at *p* < 0.05. Fisher’s exact test and chi-square test were used to evaluate the differences in categorical variables. The Kaplan–Meier method was used to estimate survival as a function of time, and survival differences were analyzed using log-rank tests. Statistical analyses were performed using the JMP 10 software from SASS (SAS Institute, Cary, NC, USA). The corresponding confidence intervals and hazard ratios were calculated using the Cox proportional hazards model.

## 3. Results

### 3.1. Patient Characteristics

Between January 2016 and December 2018 at Saitama Medical University Medical Center, 278 NSCLC patients received PD-1 blockade agents as monotherapy (nivolumab, *n* = 187; pembrolizumab, *n* = 77; atezolizumab, *n* = 14); 59 patients received first-line pembrolizumab. A total of 152 patients were excluded because of PD or SD with PFS ≤ 180 days. Finally, 126 patients who benefitted from PD-1 blockade were enrolled in our study (Figure 1). Of the 126 patients, 44 received systemic steroids because of irAEs, and 82 were not treated with systemic steroids even when any irAE occurred. We divided the 126 patients into two groups: with and without systemic steroid use. The characteristics of the patients are listed in Table 1. No significant differences in patient demographics were observed between the two groups without antimicrobial use until exacerbation or within observation period (Table 1). Median total number of ICI injection until PD or research period was 9 times (1–89) in systemic steroid use for irAE group and 21 (1–68) in the no steroid use group.

### 3.2. Efficacy and Survival Analysis

In all patients, the median PFS and OS were 14.8 months (95% confidence interval (CI): 12.3–16.4 months) and 38.4 months (95% CI: 32.0 months–not reached), respectively (Appendix A). Of the 126 patients, 87 experienced recurrence, and 48 died due to PD. The median follow-up period was 23.6 months (range: 4.7–47.7 months). The Kaplan–Meier survival curve according to the use of systemic steroids for the improvement of irAEs is shown in Figure 2. The median PFS in patients with and without the use of systemic steroids was 11.7 months (95% CI: 8.2–15.8 months) and 16.0 months (95% CI: 14.2–29.5 months), respectively, with significant difference (*p* = 0.037) (Figure 2A). No significant difference in the median OS was recognized between the two groups (35.0 vs. 41.0 months) (*p* = 0.28) (Figure 2B). In addition, we analyzed patients with systemic steroid use for reasons other than irAEs, such as palliative care and chronic obstructive pulmonary disease. Of 126 patients, 62 had no use of systemic steroids, and 64 received systemic steroids for any reason. Figure 3 shows the Kaplan–Meier survival curve for the use of systemic steroids regardless of irAEs. Patients administered systemic steroids had significantly shorter PFS than those who did not (*p* = 0.029) (Figure 3A). However, no significant difference in OS was observed between the two groups (*p* = 0.15) (Figure 3B). Exploratory univariate and multivariate analyses were conducted to identify important outcome variables (Table 2). Of the factors listed, univariate analyses demonstrated that only using systemic steroid to treat irAEs was significantly associated with longer PFS. After adjusting for multiple clinical variables, multivariate analyses revealed that using only systemic steroids alone to treat irAEs was significantly associated with improved PFS. Then, only ages above and below 70 years was significantly associated with improved OS.

### 3.3. Adverse Events

Adverse events according to the use of systemic steroids for irAEs are detailed in Table 3. Of the 82 patients who were not on systemic steroids, 70 (85.4%) experienced some irAEs. The frequency of irAEs of any grade was significantly different between the two groups of patients: with and without systemic steroid use for irAEs. In particular, the occurrence of interstitial lung disease (ILD), adrenal insufficiency, and fever was significantly predominant in patients who used systemic steroids for irAEs. On the contrary, the frequency of irAEs grade > 3, such as ILD, rash, and diarrhea, was significantly higher in patients who used systemic steroids than in those who did not. Only one patient who experienced grade 5 ILD of all patients was included in systemic steroid use group.

## 4. Discussion

In this study, we aimed to elucidate the effect of systemic steroids on irAEs due to PD-1 blockade. Compared with previous studies, our present study emphasized the clinical differences between patients with and without systemic steroid use for irAEs who gained clinical benefit from ICI treatment. To our knowledge, this is the first study to evaluate the prognostic implications of systemic steroid use during ICI treatment.

We found that systemic steroid use for irAEs affected the prognosis of patients who benefitted from treatment with PD-1 blockade. Although immunotherapy has provided a therapeutic revolution in the management of several types of malignant neoplasms, the incidence of irAEs during extensive use of PD-1/PD-L1 blockade is increasing; thus, a precise strategy using systemic steroids to treat irAEs is required. The essential endpoint of our study was to evaluate the prognostic relevance of the administration of systemic steroids to patients who could obtain clinical benefit from PD-1 blockade. The current study indicated that PFS in patients on systemic steroids was significantly shorter than that in those without steroid use, but a negative effect on OS was not observed. According to the results of univariate and multivariate, using only systemic steroids to treat irAEs was associated with prolonged PFS.

The incidence of irAEs is considered as a tumor immune reaction; therefore, they may elicit a therapeutic response to PD-1 blockade. Nevertheless, systemic steroids tend to worsen tumor immunity even if clinical benefit from PD-1 blockade treatment has been obtained. The effect of widespread use of systemic steroids on the survival benefit of PD-1 blockade needs to be studied.

Table 4 summarizes several reports focusing on the relationship between the efficacy of ICIs and systemic steroids in patients with NSCLC [11,12,15,16]. These retrospective analyses showed that the use of systemic steroids at baseline ICI treatment could affect prognosis after administration. In fact, the use of systemic steroids at baseline ICI treatment has been shown to worsen PFS in patients with NSCLC [17,18]. Moreover, the correlation between systemic steroid use at baseline ICI treatment baseline and shorter OS was observed in a large-scale cohort, including patients with melanoma, urothelial carcinoma, and NSCLC, in the USA [19]. Several studies included heterogeneous populations, such as patients continuing systemic steroids at baseline ICI treatment or those receiving steroids for brain metastasis or palliative indications, such as respiratory failure; therefore, several biases should be considered. In contrast, the current study focused on patients who gained clinical benefit from PD-1 blockade and received systemic steroids for irAEs. We attempted to reduce the heterogeneous patient bias as much as possible. However, in terms of background factors, there was a significant difference in antimicrobial use between the two groups. Other differences between the two groups may exist. Importantly, there was a difference in the number of ICIs administered in the two groups. Patients who experienced adverse events of grade 3 or higher tended to have shorter survival in univariate analysis, and the effect of less frequent administration cannot be denied.

Steroids can inhibit gene transcription, such as activator protein-1 and nuclear factor-κβ, and suppress many inflammatory pathways [20]. In general, systemic steroids are well known to reduce the expression of cytokines, such as IL-1α, IL-1β, IL-2, IL-6, IL-12, interferon-γ, tumor necrosis factor α, and granulocyte-macrophage colony-stimulating factor. Through the inhibition of cytokine production, the environment of immune-related cells was reformed, thereby altering the activity of effector CD4 and CD8 T cells. In addition, high-dose administration of systemic steroids might increase the risk of unpleasant side effects, such as endocrine disorders, hypertension, avascular necrosis of the femur, and osteoporosis. Tokunaga et al. reported that the ability of low-affinity memory CD8+ T cells was inhibited by corticosteroids [21]. Acharya et al. reported that CD8+ T cells induce dysfunction in the tumor microenvironment via endogenous glucocorticoid signaling [22]. The function of cytokines and T cells were altered by systemic steroids in patients receiving anti-PD-1/CTLA-4 antibodies, as examined by peripheral blood mononuclear cell analysis [23]. Although dexamethasone has been shown to promote the downregulation of LAG-3 expression on T cells [23], the effect of steroids in cancer patients receiving ICI treatment may be exceptional.

Systemic steroid use against serious adverse events, including irAEs, is necessary to save lives [24]. However, the available evidence regarding the dosage and administration period of systemic steroids is insufficient. Although systemic steroids can sometimes lead to serious side effects, deliberate adjustment of steroid dosage is required for patients of older age or those with comorbidities. The dose of systemic steroids should be tapered carefully in consideration for withdrawal or adrenal insufficiency [25]. A meta-analysis of the relationship between steroid use and survival in patients treated with ICIs showed that patients who received steroids showed an increased risk of progression compared with those who did not, but steroid use for irAEs did not negatively affect OS [26]. Considering the results of our study, however, systemic steroid use (PSL ≥ 10 mg, ≥2 weeks) for irAEs might negatively affect survival time. Therefore, re-administration of ICIs probably should be decently reconsidered even if grade 1 or 2 adverse events, such as lung disorders without respiratory insufficiency, skin disorders controllable by ointment, and organ derangement without aggravation, were observed during ICI treatment.

Our study had several limitations. First, our study used a retrospective approach, which may have biased the results of our study. Moreover, the use of systemic steroids for irAEs depended on the judgment of the chief physician. The timing and dosage of steroids also differed among individual patients. To correct the bias of these factors, a prospective study is necessary. Second, we defined the clinical benefit from PD-1 blockade as CR, PR, or PFS ≥ 180 days with SD (and NE), in accordance with a previous report [13]. However, whether our definition of clinical benefit is absolutely suitable may be controversial. Finally, our study lacked biological data, such as PD-L1 expression and tumor lymphocyte infiltration. Because of the limited number of tumor specimens, it was difficult to examine the expression of these markers in all tumor specimens by immunohistochemistry. It is helpful to investigate biomarkers for predicting the reduced efficacy of PD-1 blockade after systemic steroid administration. Further studies are warranted to explore the biomarkers that affect the efficacy of PD-1 blockade.

In conclusion, the patients requiring systemic steroids for irAEs due to ICI treatment exhibited shorter PFS than those without systemic steroid use. Systemic steroids might affect survival after PD-1 blockade even in patients who once acquired its clinical benefit.

## Figures and Tables

**Figure 1 jcm-10-03744-f001:**
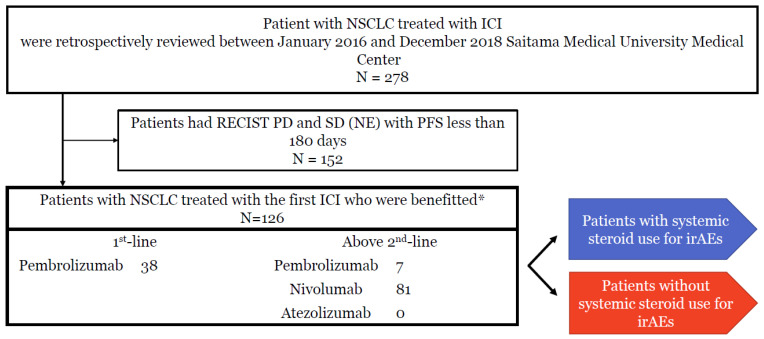
Study scheme. Among 278 patients with advanced NSCLC who received ICI monotherapy within the study period, 152 patients who had RECIST PD and SD or NE with PFS less than 180 days were excluded; 126 patients who were benefitted from ICI monotherapy were enrolled as the target population. The enrolled patients were divided into two groups: with and without systemic steroid use for irAEs. CR, complete response; PR, partial response; PD, progressive disease; NE, not evaluable. * Clinical benefit defined as RECIST CR/PR or SD and PFS of more than 180 days.

**Figure 2 jcm-10-03744-f002:**
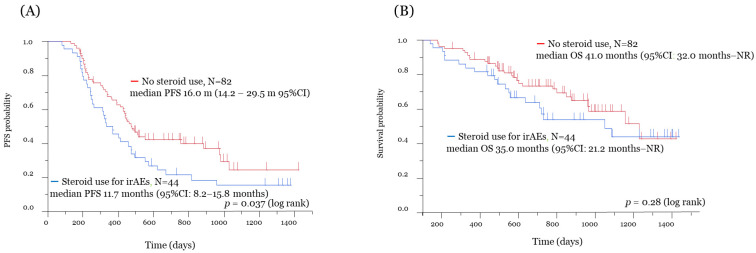
Kaplan–Meier survival analysis for the systemic steroid use for irAEs. The figures show Kaplan–Meier curves for PFS (**A**) and OS (**B**) in the systemic steroid use for irAE cohort (blue line) and the no steroid use for irAE cohort (red line). Survival analysis with ICI treatment: significant difference in PFS and no significant difference in OS. irAE, immune-related adverse event; PFS, progression-free survival; OS, overall survival; NR, not reached.

**Figure 3 jcm-10-03744-f003:**
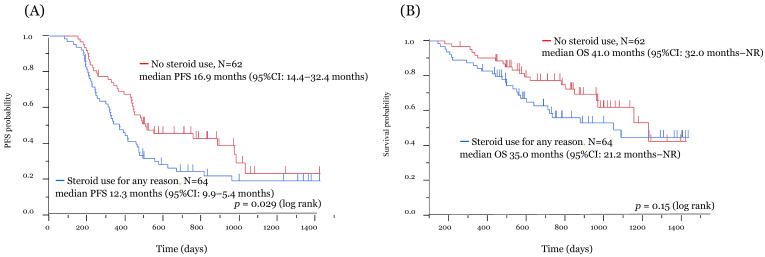
Kaplan–Meier survival analysis according to systemic steroid use for any reason. The figures show Kaplan–Meier curves for PFS (**A**) and OS (**B**) in the systemic steroid use for any reason cohort (blue line) and the no steroid use cohort (red line). Survival analysis with ICI treatment: significant difference in PFS and no significant difference in OS. irAE, immune-related adverse event; PFS, progression-free survival; OS, overall survival; NR, not reached.

**Table 1 jcm-10-03744-t001:** Patient characteristics (*n* = 126).

Variables	Systemic Steroid Use for irAEs	No Steroid Use	*p*-Value
	*n* = 44	*n* = 82	
Age			
Median (range)	70 (43–86)	70 (48–85)	
Gender			
Male/Female	38/6	69/13	0.80
PS			
0 or 1/2 or 3	35/9	73/9	0.38
Smoking history			
Yes/No	41/3	72/10	0.38
Disease stage			
III/IV/Postoperative recurrence	8/28/8	15/43/24	0.40
Histology			
Ad/Sq/Others	18/14/12	43/24/15	0.58
Driver mutation/Translocation			
EGFR/ALK/Wild type	1/0/43	5/0/77	0.65
PD-L1 (TPS)			
<1%/1–49%/>50%/Unknown	1/1/18/24	3/6/22/51	0.34
Treatment line			
1st/2nd/3rd and higher	17/20/7	21/46/15	0.33
Efficacy by ICI treatment			
CR and PR/SD/NE	28/13/3	46/31/5	0.68
Use of systemic steroid except for irAEs *			
Yes/No	5/39	15/67	0.11
Bain metastasis			
Yes/No	11/33	20/62	0.99
Re-administration ICI after irAEs			
Yes/No	27/17	46/36	0.58
Antibiotics use until PD or within observation period	27/17	28/54	0.004
Yes/No			

Abbreviations: Ad, adenocarcinoma; Sq, squamous cell carcinoma; PS, performance status; PD-L1, programmed death ligand-1; CR, complete response; PR, partial response; PD, progressive disease; NE, not evaluated; PFS, progression-free survival; TPS, tumor proportion score; ICI, Immune-checkpoint inhibitor; irAEs, immune-related adverse events. * Use of systemic steroids except for irAEs was defined as the use of a daily prednisone-equivalent dose 10 mg for at least 14 day within 30 days after ICI initiation.

**Table 2 jcm-10-03744-t002:** Associations of clinical factors with PFS and OS.

Factors	PFS						OS					
	Univariate Analysis			Multivariate Analysis			Univariate Analysis			Multivariate Analysis		
	HR	95% CI	*p* Value	HR	95% CI	*p* Value	HR	95% CI	*p* Value	HR	95% CI	*p* Value
Age <70/≧70	0.75	0.48–1.14	0.18	0.72	0.47–1.11	0.14	0.48	0.26–0.88	0.02	0.46	0.25–0.83	0.01
Gender: Male/Female	0.75	0.33–1.36	0.33				1.24	0.57–3.28	0.60			
PS 0–1/2–3	0.65	0.38–1.20	0.16	0.61	0.35–1.13	0.11	0.48	0.25–1.02	0.06			
Smoking status: Never/Current or former	0.99	0.46–1.89	0.98				0.83	0.25–2.04	0.71			
Disease stage: III or IV/Postoperative recurrence	1.25	0.78–2.11	0.35				1.53	0.79–3.24	0.22			
Histology: Ad/Sq or Others	0.82	0.53–1.24	0.34				0.66	0.36–1.17	0.16			
Driver mutation: EGFR/Wild	2.30	0.80–5.19	0.11	2.78	0.96–6.38	0.06	1.14	0.28–3.12	0.83			
Treatment line: 1st/2nd and higher	0.88	0.53–1.40	0.59				1.11	055–2.10	0.77			
Efficacy by ICI: CR and PR/SD or NE	1.24	0.81–1.90	0.32				1.03	0.58–1.85	0.93			
Antibiotics: Yes/No	0.99	0.65–1.52	0.98				1.43	0.75–2.96	0.29			
irAEs Grade≧3: Yes/No	1.45	0.91–2.27	0.12				1.86	1.01–3.32	0.04	1.94	0.99–3.73	0.05
Systemic steroid use to irAEs: Yes/No	1.56	1.01–2.38	0.04	1.67	1.07–2.57	0.02	1.38	0.76–2.44	0.28	1.06	0.54–2.01	0.86

Abbreviations: PFS, progression-free survival; OS, overall survival; PS, performance status; Ad, adenocarcinoma; Sq, squamous cell carcinoma; ICI, Immune-checkpoint inhibitor; CR, complete response; PR, partial response; PD, progressive disease; NE, not evaluated; irAEs, immune-related adverse events; HR, hazard ratio.

**Table 3 jcm-10-03744-t003:** Frequency of irAEs.

Toxicities	All Grade			Grade ≥ 3		
	Steroid Use for irAEs *n* (%)	No Steroid Use *n* (%)	*p*-Value	Steroid Use for irAEs *n* (%)	No Steroid Use *n* (%)	*p*-Value
Any irAE *1*4	44 (100)	70 (85.4)	<0.01	28 (63.6)	7 (8.5)	0.99
ILD	21 (47.7)	15 (18.3)	<0.01	7 (15.9)	1 (1.2)	<0.01
Thyroid dysfunction	8 (18.2)	7 (8.5)	0.14	0 (0)	0 (0)	0.99
Adrenal insufficiency	9 (20.5)	1 (1.2)	<0.01	3 (6.8)	1 (1.2)	0.12
Liver dysfunction *2	13 (29.5)	14 (17.1)	0.11	5 (11.4)	2 (2.4)	0.05
Renal dysfunction *3	5 (11.4)	16 (19.5)	0.32	1 (2.3)	0 (0)	0.35
Rash	15 (34.1)	24 (29.3)	0.69	3 (6.8)	0 (0)	0.04
Fever	6 (13.6)	2 (2.4)	0.02	1 (2.3)	0 (0)	0.35
Diarrhea	8 (18.2)	8 (9.8)	0.26	5 (11.4)	0 (0)	<0.01
Nervous disorder	2 (4.5)	2 (2.4)	0.61	1 (2.3)	0 (0)	0.35
Mucositis oral	2 (4.5)	3 (3.7)	0.99	0 (0)	1 (1.2)	0.99
CK	2 (4.5)	10 (12.2)	0.21	0 (0)	0 (0)	0.99
AMY	6 (13.6)	7 (8.5)	0.54	0 (0)	0 (0)	0.99
γGTP	1 (2.3)	4 (4.9)	0.66	0 (0)	0 (0)	0.99
Eosinophilia	10 (22.7)	19 (23.2)	0.99	1 (2.3)	0 (0)	0.35

Abbreviations: irAEs, immune-related adverse events; ILD, interstitial lung disease; CK, creatin kinase; AMY, amylase; γGTP, γ-glutamyl trans peptidase. *1 We listed the irAEs with a frequency of more than 3% in either group. *2 Liver dysfunction was defined as the elevation of aspartate aminotransferase (AST) or alanine aminotransferase (ALT). *3 Renal dysfunction was defined as an increase in creatinine (Cr). *4 Several patients experienced overlapping immune-related adverse events.

**Table 4 jcm-10-03744-t004:** Review of literatures reporting a relationship between systemic steroid use and ICI treatment in NSCLC.

First Author (Reference)	Eligible Subjects Who Received ICI	Number of Patients			ICI Regimens	Reason of Steroid Initiation	Timing of Steroid Initiation with PSL >10 mg	Survival after ICI Initiation	
		All Patients	Patients with Steroid Use	Patients without Steroid Use				PFS (Steroid Use vs. No Steroid Use) (*p*-Value)	OS (Steroid Use vs. No Steroid Use) (*p*-Value)
Arbour *1 [11]	All patients (cohort 1) All patients (cohort 2)	455 185	53 37	402 148	Nivolumab Pembrolizumab Atezolizumab Durvalumab	Palliative or brain metastasis	ICI initiation	1.7 m vs. 1.8 m (*p* < 0.001) 1.9 m vs. 2.6 m (*p* = 0.001)	3.3 m vs. 9.4 m (*p* < 0.001) 5.4 m vs. 12.1 m (*p* < 0.001)
Fuca [12]	All patients	151	35	116	Anti-PD-(L)1 /Anti PD-L1 + anti-CTLA-4	Palliative or brain metastasis	Within 28 days after ICI initiation	1.98 m vs. 3.94 m (*p* = 0.003)	4.83 m vs. 15.14 m (*p* < 0.01)
Ricciuti [15]	All patients	650	93	557	Anti-PD-1/PD-L1 Anti-CTLA-4	Palliative or brain metastasis, etc.	Within 24 h after ICI initiation	2.0 m vs. 3.4 m (*p* = 0.01)	4.9 m vs. 11.2 m (*p* < 0.001)
Scott [16]	All patients	210	66	144	Nivolumab	Palliative or brain metastasis, COPD, irAEs	Within 30 days after ICI initiation	N/A	4.3 m vs. 11.0 m (*p* = 0.006)
Present study	Patients with clinical benefit from ICI	126	44	82	Nivolumab Pembrolizumab	irAEs	All period of ICI treatment	11.7 m vs. 16.0 m (*p* = 0.037)	35.0 m vs. 41.0 m (*p* = 0.28)

Abbreviations: Retro, retrospective; PD-(L)1, programmed cell death—(ligand)1; CTLA-4, cytotoxic T-lymphocyte associated antigen-4; ICI, immune check point inhibitor; PSL, prednisolone; COPD, chronic obstructive pulmonary disease. CR, complete response; PR, partial response; N/A, not applicable. *1 This report disclosed two individual analyses from two separate institutions.

## Data Availability

The data presented in this study are available on motivated request to the corresponding author.

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
