# Peer review of "Effect of Systemic Steroid Use for Immune-Related Adverse Events in Patients with Non-Small Cell Lung Cancer Receiving PD-1 Blockade Drugs"

_jcm, 2021, doi:10.3390/jcm10163744_

Round 1

Reviewer 1 Report

General comments

I think this work is very interesting and clinically relevant. The paper is well written, and the design quite well done.

However, I noticed several limitations which should be corrected. I saw some bias and I think the authors must put shades in their conclusions.

I think this is important to discuss the fact that there were significantly more severe side effects in the the group of patients who received steroids. They also should add in the manuscript if there is a difference between the two groups concerning the number of immunotherapy injections. Less treatment injections could explain the PFS difference.

I also think that it would be interesting to do univariate and multivariate analyses to show that steroid administration is responsible of the difference of survival.

I also think that it could be interesting to add or at least to discuss the other factors of poor response to immunotherapy which are described in the literature such as antibiotics (PMID: 34323149).

The number of deaths by immune toxicity is not described and might also change the interpretation of the results.

The other comments are detailed below.

To summarize, this article is interesting but, in my opinion, it has to be improved to be published in Journal of Clinical Medicine. Major modifications are required before publication in Journal of Clinical Medicine.

Detailed comments:

Abstract: I think that the authors might precise that there were more side effects in the group treated by steroids.

Methods:

  • I am not sure to understand what NE status is. If the response is not evaluable, why do you keep these patients in your analysis?
  • Line 92: this is written: “with and without systemic steroid use AFTER cessation of PD-1 blockade”. Why AFTER cessation? I understood it was during the anti cancer therapy. May you clarify this point?
  • Line 99-101: is there any patients who received concomitant chemotherapy?

Results:

  • Line 136: When the authors described the 59 patients, they should precise that they are among the 278 patients and that only 59 patients received the treatment as monotherapy first line.
  • Paragraph: 3.2. I really think that the number of received injections has to be clearly written. I also think that univariate and multivariate analysis should be done to be sure the the steroids are responsible of the difference of survival. In fact, there are a lot of factors which can explain the results (PS, age, line of treatment, grade of the side effects, number of received injections…)
  • Paragraph 3.3: I really do not understand why there were only 85.4% (and not 100%) of patients who experienced side effects in the group with no steroid. I thought it was a criterion of inclusion!!! The authors have to clarify this point which seems to be a big methodological issue… In my opinion the patients with no side effect have to be removed of the analysis

Discussion:

As I wrote before, I really think that limitations and bias of the study have to be discussed more in details to put shades in the conclusions:

  • Higher rate of side effects and > or = grade 3 side effects
  • Others causes of less efficiency (such as antibiotics).
  • Plus, as previously written, some results are lacking such as univariate and multivariate analyses and number of received injections in each group.
  • The imprecisions concerning methods have to be resolved or discussed: why do we have patients with no side effect in the non steroids groups? What does NE mean? When did the patients receive steroids?

Author Response

I have revised it based on your precise and excellent suggestions. I have collected data on antimicrobial use and added it to the background table. I have performed univariate and multivariate analyses, including antimicrobial use, and added a new Table. I believe that your valuable suggestions have helped us to turn this into a high quality paper. I would like to ask you to review the paper again.

Detailed comments:

Abstract: I think that the authors might precise that there were more side effects in the group treated by steroids.

→Thank you for your remarks. I've added to abstract. (”More side effects were observed in the group treated by steroids”)

Methods:

  • I am not sure to understand what NE status is. If the response is not evaluable, why do you keep these patients in your analysis?

→Although our study included NE cases, they could gain clinical benefit from PD-1 blockade. Actually, it may be difficult to evaluate the clinical response of ICI treatment by only RECIST. We think that NE cases with therapeutic benefit are eligible.

  • Line 92: this is written: “with and without systemic steroid use AFTER cessation of PD-1 blockade”. Why AFTER cessation? I understood it was during the anti cancer therapy. May you clarify this point?

→ Thanks for pointing that out. I have removed “AFTER cessation of PD-1 blockade” from that sentence.

  • Line 99-101: is there any patients who received concomitant chemotherapy?

→ The patients treated with concomitant chemotherapy and Immune check point inhibitor were not included in this study. Research period of this study is between January 2016 and December 2018 prior to approval for administration in combination with chemotherapy in Japan.

Results:

  • Line 136: When the authors described the 59 patients, they should precise that they are among the 278 patients and that only 59 patients received the treatment as monotherapy first line.

Research period of this study is between January 2016 and December 2018, so fifty-nine patients were actually treated with pembrolizumab as first-line therapy.

  • Paragraph: 3.2. I really think that the number of received injections has to be clearly written. I also think that univariate and multivariate analysis should be done to be sure the the steroids are responsible of the difference of survival. In fact, there are a lot of factors which can explain the results (PS, age, line of treatment, grade of the side effects, number of received injections…)

Thank you for your valuable suggestions. I've appended ”Median total number of ICI injection until PD or research period was 9 times (1-89) in systemic steroid use for irAE group and 21 (1-68) in no steroid use group” to 3.1. Patient characteristics. 

We do not believe that the more immune checkpoint inhibitors a patient receives, the more a patients will benefit. There have been many reports of patients remaining progression-free after only a few doses. We believe that the appropriate number of doses may differ for each individual case. We believe that the number of received injections may vary depending on the individual case. In cases where serious adverse events occur, it may be assumed that the immune response is enhanced. Therefore, we believe that the number of received injections does not have a significant effect on survival.

  • Paragraph 3.3: I really do not understand why there were only 85.4% (and not 100%) of patients who experienced side effects in the group with no steroid. I thought it was a criterion of inclusion!!! The authors have to clarify this point which seems to be a big methodological issue… In my opinion the patients with no side effect have to be removed of the analysis

The current study investigates the impact of steroid administration on survival in a limited selection of patients who have gained benefit from immune checkpoint inhibitor therapy.  Therefore, patients who did not experience any irAEs were included in the target population if they benefited from ICI therapy.

Discussion:

As I wrote before, I really think that limitations and bias of the study have to be discussed more in details to put shades in the conclusions:

  • Higher rate of side effects and > or = grade 3 side effects

→ Thanks for your astute remarks. The only one patient who experienced grade 5 ILD of all patients was included in systemic steroid use group. I described it in Results 3.3.

  • Others causes of less efficiency (such as antibiotics). 

Thanks for the great suggestion. Additional chi-square test was performed and added to the body text as Table 1. We performed univariate and multivariate analyses of antimicrobial use, which showed a no significant difference between the two groups.

  • Plus, as previously written, some results are lacking such as univariate and multivariate analyses and number of received injections in each group.

Thanks for the great suggestion. Additional univariate and multivariate analyses were performed and added to the body text as new Table 2.

  • The imprecisions concerning methods have to be resolved or discussed: why do we have patients with no side effect in the non steroids groups? What does NE mean? When did the patients receive steroids?

   →The original concept of this study was to investigate the effect of steroid administration only in patients who achieved a survival benefit from immune checkpoint inhibitor therapy. Therefore, we decided to include patients without adverse events or with NE who have benefited from immune checkpoint inhibitor therapy in this study.

Reviewer 2 Report

The paper “Effect of systemic steroid use for immune-related adverse events in patients with non-small cell lung cancer receiving  PD-1 blockade drugs” is quite interesting but rumbling and difficult to read. I have several comments:
1) Abstract. Results. Please ameliorate this paragraph.2) Introduction. Systemic steroids, acting as anti-inflammatory agents, were administered to patients with autoimmune disorders to suppress the immune response. Generally, systemic steroids have been suggested to reduce phagocytic activity; inhibit leukocyte migration; reduce interleukin (IL)-52 1, IL-6, IL-8, IL-12, and tumor necrosis factor-α; and impair T cell activity and differentia-53 tion.ï¼»9,10ï¼½Please improve this paragraph and add these references:a) Ruaro B, Salton F, Braga L, Wade B, Confalonieri P, Volpe MC, Baratella E, Maiocchi S, Confalonieri M. The History and Mystery of Alveolar Epithelial Type II Cells: Focus on Their Physiologic and Pathologic Role in Lung. Int J Mol Sci. 2021 Mar 4;22(5):2566. doi: 10.3390/ijms22052566.b) Meduri GU, Annane D, Confalonieri M, Chrousos GP, Rochwerg B, Busby A, et al. Pharmacological principles guiding prolonged glucocorticoid treatment in ARDS. Intensive Care Med. 2020 Dec;46(12):2284-2296. doi: 10.1007/s00134-020-06289-8.

2) Introduction. L 77- 80. On the basis of this background, we retrospectively assessed the difference in efficacy  and survival in patients with NSCLC on PD-1 blockade agents, who were administered  steroids to manage irAEs. The aim of this study was to elucidate whether systemic steroids administered to manage irAEs affect the benefits of PD-1 blockade in patients with metastatic or recurrent NSCLC. Please improve the description of study aim.

3) Methods. L98-103.  2.2. Treatment and adverse events Pembrolizumab, nivolumab, and atezolizumab were intravenously administered: 3  mg/kg or 240 mg/day every 2 weeks, 200 mg/day every 3 weeks, and 1200 mg/day every  3 weeks, respectively. Complete blood cell count, differential count, routine chemistry  measurements, physical examination, and toxicity levels were evaluated through medical  examination by physicians. Toxicities were graded according to the CTCAE version 5.0. In accordance with the judgment of each physician, ICI treatment was repeated until disease progression, appearance of severe toxicity, or patient’s refusal for treatment. Please improve the description of the different treatment regimens.

4) Methods. Please ameliorate the ethics committee approval information.

5) 2.4. Statistical analysis L124-128. Statistical analyses were performed at p <0.05. Fisher’s exact test, Chi-square test, and U-Mann Whitney test were used to examine the association between the two groups with or without systemic steroid use for irAEs. The Kaplan–Meier method was used to estimatesurvival as a function of time, and survival differences were analyzed using log-rank tests.  Statistical analyses were performed using the JMP 10 software from SASS. The corresponding confidence intervals and hazard ratios were calculated using the Cox proportional hazards model. Please this section must be improve.

6) Statistical analysis. I think that is necessary a multivariate analysis to support the data.

7) 3. Results 3.1. Patient characteristics L 133 -  136. Between January 2016 and December 2018 at Saitama Medical University Medical  Center, 278 NSCLC patients received PD-1 blockade agents as monotherapy (nivolumab, n=187; pembrolizumab, n=77; atezolizumab, n=14); 59 patients received first-line pembrolizumab. A total of 152 patients were excluded because of PD or SD with PFS ≤180  days. Finally, 126 patients who benefitted from PD-1 blockade were enrolled in our study  (Figure. 1). Of the 126 patients, 44 received systemic steroids because of irAEs, and 82  were not treated with systemic steroids even when any irAE occurred. We divided the 126  patients into two groups: with and without systemic steroid use. The different groups of patients are unbalanced, I think that it could influence the results. Do you have any comments?

8) 4. Discussion. L 275-278. The timing and dosage of steroids also differed among individual patients. To correct the bias of these factors, a prospective study is necessary. Second, we defined the clinical benefit from PD-1 blockade as CR, PR, or PFS ≥180 days with SD (and NE) in accordance with a previous report.Do you think that the different therapeutic regimens may have influenced the results?

9) The number of references must be revised in the text and in the tables because there are some errors.

Author Response

I have revised it based on your precise and excellent suggestions. I have performed univariate and multivariate analyses and added a new Table. I believe that your valuable suggestions have helped us to turn this into a high quality paper. I would like to ask you to review the paper again.

1) Abstract. Results. Please ameliorate this paragraph.

Thanks for pointing that out. I have revised and added the description of the target group and adverse events.

2) Introduction. Systemic steroids, acting as anti-inflammatory agents, were administered to patients with autoimmune disorders to suppress the immune response. Generally, systemic steroids have been suggested to reduce phagocytic activity; inhibit leukocyte migration; reduce interleukin (IL)-52 1, IL-6, IL-8, IL-12, and tumor necrosis factor-α; and impair T cell activity and differentia-53 tion.ï¼»9,10ï¼½Please improve this paragraph and add these references:a) Ruaro B, Salton F, Braga L, Wade B, Confalonieri P, Volpe MC, Baratella E, Maiocchi S, Confalonieri M. The History and Mystery of Alveolar Epithelial Type II Cells: Focus on Their Physiologic and Pathologic Role in Lung. Int J Mol Sci. 2021 Mar 4;22(5):2566. doi: 10.3390/ijms22052566.b) Meduri GU, Annane D, Confalonieri M, Chrousos GP, Rochwerg B, Busby A, et al. Pharmacological principles guiding prolonged glucocorticoid treatment in ARDS. Intensive Care Med. 2020 Dec;46(12):2284-2296. doi: 10.1007/s00134-020-06289-8.

Thanks for the very helpful suggestions about mechanism of systemic steroids. I have rewritten and quoted it as you suggested.

2) Introduction. L 77- 80. On the basis of this background, we retrospectively assessed the difference in efficacy and survival in patients with NSCLC on PD-1 blockade agents, who were administered steroids to manage irAEs. The aim of this study was to elucidate whether systemic steroids administered to manage irAEs affect the benefits of PD-1 blockade in patients with metastatic or recurrent NSCLC. Please improve the description of study aim.

Thanks for pointing that out. I've rewritten it for clarity.

3) Methods. L98-103.  2.2. Treatment and adverse events Pembrolizumab, nivolumab, and atezolizumab were intravenously administered: 3 mg/kg or 240 mg/day every 2 weeks, 200 mg/day every 3 weeks, and 1200 mg/day every 3 weeks, respectively. Complete blood cell count, differential count, routine chemistry measurements, physical examination, and toxicity levels were evaluated through medical examination by physicians. Toxicities were graded according to the CTCAE version 5.0. In accordance with the judgment of each physician, ICI treatment was repeated until disease progression, appearance of severe toxicity, or patient’s refusal for treatment. Please improve the description of the different treatment regimens.

Thanks for pointing that out. I have already proofread the English.

4) Methods. Please ameliorate the ethics committee approval information.

→ Thank you for pointing this out. I have added the exact date to the ethics committee approval information in “2.1. Study design and subjects”.

5) 2.4. Statistical analysis L124-128. Statistical analyses were performed at <0.05. Fisher’s exact test, Chi-square test, and U-Mann Whitney test were used to examine the association between the two groups with or without systemic steroid use for irAEs. The Kaplan–Meier method was used to estimate survival as a function of time, and survival differences were analyzed using log-rank tests.  Statistical analyses were performed using the JMP 10 software from SASS. The corresponding confidence intervals and hazard ratios were calculated using the Cox proportional hazards model. Please this section must be improve.

Thanks for pointing that out. I've rewritten it.

6) Statistical analysis. I think that is necessary a multivariate analysis to support the data.

Thank you for your sharp remarks. I have added the results of univariate and multivariate analysis as new Table2.

7) 3. Results 3.1. Patient characteristics L 133 - 136. Between January 2016 and December 2018 at Saitama Medical University Medical Center, 278 NSCLC patients received PD-1 blockade agents as monotherapy (nivolumab, n=187; pembrolizumab, n=77; atezolizumab, n=14); 59 patients received first-line pembrolizumab. A total of 152 patients were excluded because of PD or SD with PFS ≤180 days. Finally, 126 patients who benefitted from PD-1 blockade were enrolled in our study (Figure. 1). Of the 126 patients, 44 received systemic steroids because of irAEs, and 82 were not treated with systemic steroids even when any irAE occurred. We divided the 126 patients into two groups: with and without systemic steroid use. The different groups of patients are unbalanced, I think that it could influence the results. Do you have any comments?

→ You are right, I think there are other unbalanced factors besides the assessment of group differences between the two groups in table1. In this revision, we added the presence or absence of antimicrobial use to the group comparison, and the chi-square test showed a significant difference in the presence or absence of antimicrobial use until disease progression.

8) 4. Discussion. L 275-278. The timing and dosage of steroids also differed among individual patients. To correct the bias of these factors, a prospective study is necessary. Second, we defined the clinical benefit from PD-1 blockade as CR, PR, or PFS ≥180 days with SD (and NE) in accordance with a previous report. Do you think that the different therapeutic regimens may have influenced the results?

→ The influence of steroid dosage and timing of intervention on survival is still unknown. To explore the effects of steroids in more depth, we are currently conducting research using peripheral blood mononuclear cells. 

In terms of PD-1 inhibitors versus PD-L1 inhibitors, I think the impact on survival will be minimal, but I think there will be an impact on if immune checkpoint inhibitors are introduced in first-line therapy versus later intervention.

9) The number of references must be revised in the text and in the tables because there are some errors.

Thank you for reading carefully. I have corrected it.

Round 2

Reviewer 1 Report

Thank you very much for your clear reply.

I have only few additional comments.

Once again, I think that it would interesting to discuss the fact that there were more > or = grade 3 side effects in the no steroids group. I would also add it in the univariate and multivariate analysis as a variable. It could really be a bias. So, I believe this is important to discuss it and/or include it in your univariate and multivariate analysis.

Finally, I would add in the discussion a sentence about the difference of the number of received injections between the two groups. I understand your point of view about the number of doses which may differ for each individual case. However, some studies do not support your point of view such as the results of CheckMate 153. So, it could be a bias and should be clearly discussed.

Author Response

Thank you for your excellent point. In this revision, we added serious adverse event occurrence to univariate and multivariate analysis. There was no significant difference in the multivariate analysis, but univariate analysis showed that those with grade 3 or higher adverse events had a significantly shorter overall survival correlation.

 I agree with you and have learned from the Check Mate153 study that termination at 1 year may have a negative impact on prognosis. Dose interruptions may well be a factor in poor prognosis, and those with serious adverse events who would have received fewer doses tended to have shorter survival in univariate analysis. The following document has been added to Discussion.

Importantly, there was a difference in the number of ICIs administered in the two groups.  Patients who experienced adverse events of grade 3 or higher tended to have shorter survival in univariate analysis, and the effect of less frequent administration cannot be denied.”

Reviewer 2 Report

The manuscript has been improved and corrected as required.

Author Response

I would like to thank you for taking time out of your busy schedule to review my work.